# Contribution of Tibialis Anterior in Sit-to-Stand Motion: Implications for Its Role in Shifting the Center of Pressure Backward

**DOI:** 10.3390/jfmk10020156

**Published:** 2025-05-01

**Authors:** Hiroki Hanawa, Taku Miyazawa, Keisuke Hirata, Keisuke Kubota, Tsutomu Fujino

**Affiliations:** 1Department of Rehabilitation, Faculty of Health Science, University of Human Arts and Sciences, 354-3 Shinshoji-Guruwa, Ota-aza, Iwatsuki-ku, Saitama-shi 339-8555, Saitama, Japan; 2Department of Rehabilitation, Faculty of Health Sciences, Tokyo Kasei University, 2-15-1 Inariyama, Sayama-shi 350-1398, Saitama, Japan; 3Research and Development Center, Saitama Prefectural University, 820 San-Nomiya, Koshigaya-shi 343-8540, Saitama, Japan

**Keywords:** tibialis anterior, sit-to-stand, center of pressure, functional anatomy, electromyography

## Abstract

**Background:** The role of tibialis anterior activity in sit-to-stand motion is unclear. We hypothesized that contraction of the tibialis anterior would slightly lift the forefoot and shift the center of pressure backward. **Objectives:** The objective of this study was to clarify this movement and its role in tibialis anterior activity. **Methods:** Ten healthy adults performed the sit-to-stand motion. Cross-correlation coefficients among tibialis anterior activity, shank inclination angle, and center of pressure were calculated. Whole-body joint moments were simulated when the center of pressure varied within the foot. The angle of the ground reaction force during seat-off was calculated. **Results:** The center of pressure moved backward in all trials for all participants. The mean lag time for peak cross-correlation coefficients between the tibialis anterior and shank tilt and between the tibialis anterior and center of pressure was 0.37 and 0.13 s, respectively. Simulating the center of pressure forward resulted in greater whole-body joint moments than those measured (mean 1.88 times). The ground reaction forces were nearly perpendicular to the floor. **Conclusions:** From the perspective of temporal synchrony, tibialis anterior activity significantly contributed to the backward shift of the center of pressure. The center of pressure shift minimized the force exerted by the entire body.

## 1. Introduction

A well-known physical characteristic of humans is the bipedal stance [1]; hence, the muscles that extend the legs are often highlighted. The triceps surae form the calf as an extensor of the ankle joint. In the standing position, extension of the ankle joint implies standing on the toes [2]. The tibialis anterior, located on the shin, maintains balance by shifting the center of pressure toward the heel [3].

However, the role of the tibialis anterior in the sit-to-stand motion, an important prerequisite for bipedal standing, remains unclear. The tibialis anterior is attached to the anterior surface of the shank and the dorsal surface of the forefoot [4]. Contraction of the tibialis anterior draws its attachments together. In other words, it produces a force that tilts the shank forward on one side and lifts the forefoot toward the other side (Figure 1). Electromyographic (EMG) and kinematic studies have shown that tibialis anterior activity and forward shank tilt occur during the sit-to-stand motion [5,6,7]. Therefore, it has been hypothesized that contraction of the tibialis anterior causes forward shank tilt [5].

However, the shank is heavier than the foot [8,9]. If the tibialis anterior contracts, the force that attracts both should result in the forefoot lifting first. Observing the sit-to-stand motion, the ankle angle does not significantly change (Figure 1) [5,6,7]. However, it may be lifted sufficiently to move the center of pressure to the heel, as in the bipedal stance.

We hypothesized that tibialis anterior activity shifts the center of pressure backward. The purpose of this study was to determine the role of the tibialis anterior in sit-to-stand motion from the perspective of synchrony between tibialis anterior activity and the resulting motion (Figure 1). The role of the center-of-pressure shift was also examined from the perspective of whole-body force minimization and balance.

## 2. Materials and Methods

### 2.1. Participants

Ten healthy adults participated in this study. To account for differences by age, eight younger and two older participants were included (Table 1). The study was conducted in accordance with the ethical principles of the Declaration of Helsinki, and written informed consent was obtained from each participant. This study was approved by the Ethics Review Committee of Saitama Prefectural University.

### 2.2. Procedure

Each participant performed the sit-to-stand motion at a comfortable speed. Three trials were conducted after completing several preliminary trials. Data from each trial were included in the analysis without averaging. This was to avoid extracting trends common to each participant and to address phenomena that were consistent within and across participants. To eliminate the influence of body height on movement patterns during the sit-to-stand task, seat height was individually adjusted [10]. The seat height corresponded to each participant’s patella height so that each foot was placed slightly behind the knee joint (Figure 1). This procedure minimized variability in joint movements and muscle activity between participants, thereby enhancing comparability across participants. Participants were instructed: “Both arms should be lowered next to the trunk. Arms should not touch the chair or thighs”.

### 2.3. Data Collection

During the sit-to-stand motion, a camera motion capture system (Vicon Motion Systems, Oxford, UK) was used at 100 Hz to measure the 3D positions of 39 passive retroreflective markers. Four force plates (Kistler Instrumente AG, Winterthur, Switzerland) were used at 1000 Hz, with two placed under the participants’ feet and the others under the platform. In addition, a surface EMG system (Delsys Trigno, Boston, MA, USA) was used at 1000 Hz to collect EMG data from the tibialis anterior of both legs. All data were synchronized using Vicon Nexus v2.7 and stored on a disk for offline analysis.

### 2.4. Data Processing

A human body model was constructed to calculate segment and joint angles. A plug-in gait model implemented in the camera motion capture system was used. The center of pressure was calculated based on the reaction forces and moments obtained from each force plate [11].

The measured data were extracted during the sit-to-stand motion [12]. The starting point was defined as the point at which the angular velocity of either joint began to change continuously. The endpoint was defined as the point at which all the joint angular velocities reached 0 rad/s. The seat-off timing was defined as the time at which the reaction force from the platform reached 0 N.

Seat-off is a critical event in the sit-to-stand motion from a mechanical perspective [7,13,14]. The center of mass, which was on the seat at the start of the motion, moved onto the feet during seat-off. When the joint moments are at their peak, the transition to the bipedal stance occurs.

### 2.5. Correlation and Synchrony Between Tibialis Anterior, Shank Tilt, and Center of Pressure

In this study, cross-correlation coefficients were employed to assess the synchronization between muscle activity and movements. This method enables the evaluation of temporal relationships between the two signals by quantifying the degree and timing of their correlation. Following the approach of a previous study [15], the lag time corresponding to the peak cross-correlation value was used to determine the temporal offset between the signals. A small lag time indicates strong temporal synchronization, and a consistent lead–lag pattern provides insight into the sequence of muscle activation and resulting motion.

Cross-correlation coefficients were calculated for tibialis anterior activity and shank inclination angle, as well as for tibialis anterior activity and center of pressure. Synchronization was verified by calculating the lag times of the peak correlation. Considering the direction of action of the tibialis anterior, the correlation between tibialis anterior activity and shank inclination angle should be positive and the correlation between the tibialis anterior activity and the center of pressure should be negative. This is because the angle and position of the sagittal plane have direction, with forward considered positive and backward considered negative.

EMG activity and movement are different physical quantities. Because these units are different and the baselines are not aligned, the waveforms cannot be compared. Therefore, we first converted each parameter to its rate of change (time differentiation) and aligned the baseline to zero. The maximum value was normalized to one. The center of pressure was normalized with the minimum value set to −1, since it was hypothesized to move backward. Each parameter was normalized because the magnitude of the measured muscle activity and the resulting movement would differ between participants. There is a distance from the electrodes to the muscle fibers where the action potentials are detected, and the thickness of the fat layer varies across participants [16].

### 2.6. Center of Pressure and Its Relationship to Whole-Body Joint Moments

The whole-body joint moments were simulated when the center of pressure moved within the foot during seat-off. Seat-off timing was employed because it occurs when the joint moments reach their peak [7,13,14]. The foot length was normalized to 100% from the heel to the proximal phalanx of the big toe.

By incorporating ground reaction forces into the constructed human body model, 2D (sagittal plane) joint moments were calculated for the ankle, knee, hip, and waist [9,11,17]. All absolute values were summed, and whole-body joint moments were considered.

### 2.7. Relative Position of Center of Pressure and Mass and Angle of Ground Reaction Force

The center of pressure at seat-off was converted to the difference from the horizontal center of mass. The variability was compared with the center of pressure from the heel. The relative position of the center of pressure and mass is a common measure of balance used in related studies [18,19,20].

The angle to the floor at seat-off was calculated for the ground reaction force vector, starting from the center of pressure. The ground reaction force effectively lifted the center of mass upward if it was perpendicular to the floor. Previous studies have also emphasized the importance of the vertical ground reaction force as a parameter related to motion speed and success [14,21,22].

## 3. Results

The foot center of pressure was displaced backward in the first half of the motion in all trials for all participants.

The mean peak cross-correlation coefficients between tibialis anterior activity and shank inclination angle (both converted to rate of change) were 0.85 (Table 2). The mean peak cross-correlation coefficients between tibialis anterior activity and center of pressure (both converted to rate of change) were −0.90 (Table 2). The lag times to peak values were 0.37 (range 0.05–0.53) and 0.13 (range 0.01–0.26) s, respectively (Table 2). The first trial for Participant 1 is shown in Figure 2. The trials for all the participants are shown in Figure A1 and Figure A2.

The simulated results of the whole-body joint moments when the center of pressure is displaced within the foot are shown in Figure 3 and Table 3. For all participants and trials, the forward displacement of the center of pressure resulted in greater whole-body joint moments than those measured (mean 1.88 times). In contrast, when displaced backward, they remained largely unchanged from the measured values (mean 0.92 times).

The center of pressure at seat-off is shown in Figure 4. The conversion to the difference between the center of mass and pressure is also shown in Figure 4. The variability in the center of pressure was greater when converted to a position relative to the center of mass.

The angle of the ground reaction force to the floor surface at seat-off is shown in Figure 4. It was consistently close to 90° (i.e., perpendicular to the floor) in all trials for all participants.

## 4. Discussion

From the perspective of temporal synchrony, it was clear that the tibialis anterior contributed more directly to the backward shift of the foot center of pressure than to the forward shank tilt (Figure 2, Table 2). In addition, whole-body joint moments were smaller when the center of pressure was backward (Figure 3). The relative positions of the center of pressure and mass at seat-off varied, making it difficult to interpret this as simply a contribution to the sit-to-stand balance (Figure 4). However, the ground reaction force vector starting from the center of pressure was consistently close to the vertical direction (Figure 4). The center of pressure was involved in both force minimization and balance in the sense that it effectively produced a force that carried the center of mass upward (i.e., a smaller force to slip).

### 4.1. Tibialis Anterior Activity and Center-of-Pressure Shift

Tibialis anterior activity and the center-of-pressure shift were strongly correlated with a short time delay (Figure 2, Table 2). As hypothesized, this result suggests that tibialis anterior activity moves the center of pressure backward. This is a new fact that contradicts previous studies directly linking tibialis anterior activity to forward shank tilt [5].

The resulting force or motion is delayed by the electrical activity of the muscle. This is called electromechanical delay [23]. This delay is estimated to be approximately 0.03–0.1 s, but it varies depending on the muscle, movement, and measurement technique [23,24,25,26]. For example, in a previous study on the tibialis anterior, the electromechanical delay of force ranged from 0.11 to 0.36 s for isometric contractions of 1% to 10% of maximal muscle force [27]. Another study also found a median (interquartile range) electromechanical delay of moment of 0.11 (0.19) s for ramp isometric contraction tasks up to 80% [28]. In the present study, neither the forward shank tilt nor the center-of-pressure shift deviated significantly from the mean electromechanical delay at which the cross-correlation coefficient was maximal or minimal (Table 2). However, the electromechanical delay in the forward shank tilt was longer. The maximum value of 0.53 s deviates significantly from the results of previous studies and is not realistic.

During sit-to-stand motion, the tibialis anterior is active before the other major leg muscles [6,7]. The only muscles that were active before seat-off were the muscles around the trunk, where most of the body movement occurred, and the tibialis anterior. Immediately after seat-off, other leg muscle activities and forward shank tilt were maximal. That is, the forward shank tilt is maximized after proximal muscle activity and body movement. Therefore, these muscle activities and body movements may be included as sources of forward shank tilt.

The tibialis anterior contraction did not significantly lift the forefoot. However, its relatively small force moved backward at the center of pressure. Some previous studies have considered the role of the tibialis anterior as “stabilizing the ankle joint”, focusing on the fact that there was no significant change in ankle angle or muscle length [5,29,30]. However, the center-of-pressure shift that we identified here is closely related to the ground reaction force and the center of mass [19,31], and needs to be described in relation to the whole body.

### 4.2. Effect of Center-of-Pressure Shift on Minimizing Whole-Body Joint Moments

The sit-to-stand motion begins with whole-body joints in a flexed position and leads to straight standing. Therefore, large extension moments are required around all joints of the body [32]. In a study that simulated the minimum moment required for sit-to-stand motion in 5,527,125 different postures, the main limiting factor for success was the magnitude of the hip and knee extension moments [33]. The required ankle extension moments were relatively trivial and exhibited high variability among the successful simulations. The authors emphasized the importance of hip and knee extension moments as a mechanical aspect of the sit-to-stand motion because of their amplitude and consistency.

We agree with this point. However, we discovered a new fact: contraction of the tibialis anterior, the resulting small ankle dorsiflexion moment, and the large backward shift of the center of pressure contributed significantly to the minimization of the whole-body joint moments. The results of this study show that simulating the center of pressure on the toe requires a mean of 1.88 (maximum 2.38) times the measured joint moment (Table 3). There were simulations in which the moments required were smaller than the measured values, but the mean was 0.92 (minimum 0.78), with little variability relative to the maximum values (Table 3). This is important from a motion-control perspective. When the center of pressure was simulated over a range of approximately 40% behind the foot length, there was no significant variability in whole-body joint moments (Figure 3). The center of pressure does not have to be near the heel or ankle joints. The fact that strict control is not required for dynamic motion is practical and has significant advantages.

### 4.3. Center of Pressure and Sit-to-Stand Balance

In the standing position, the tibialis anterior moves backward at the center of pressure, in contrast to the action of the triceps surae, which maintains balance [2,3]. From a mechanical perspective, balance can be interpreted as a small oscillation of the centers of pressure and mass toward each other at the coincidence point [31]. When the support surface transitions to the feet while standing up from the seat (i.e., seat-off), the center of mass has a forward velocity and is therefore located behind the heel [13]. However, we hypothesized that contraction of the tibialis anterior may shift the center of pressure backward, thereby welcoming the center of mass and preserving sit-to-stand balance.

Contrary to this hypothesis, the relative positions of the center of pressure and mass showed more variability than the position of the center of pressure from the heel (Figure 4). Perhaps because the sit-to-stand motion is more dynamic than standing, the balance should have been described by taking velocity information into account. Yoshioka et al. simulated the minimization of whole-body joint moments during a slow sit-to-stand motion, where the positions of the center of pressure and mass coincided [33]. However, the role of center-of-pressure shift in sit-to-stand balance is not clear from the parameters often used in balance studies [18,19,20].

Here, the ground reaction force should be extended perpendicularly to the floor surface to support the body. This is because the sit-to-stand motion carries the body upwards. In addition, if it is closer to the perpendicular side, the shear force is less and the frictional force is greater, making it more difficult to slip.

The angle between the ground reaction force vector and floor surface was calculated from this perspective. This angle was consistently close to the vertical angle (Figure 4). Whole-body extensor activity pushed the floor vertically, effectively carrying the center of mass upward. We found that the center of pressure was adjusted such that the motion could be accomplished vertically with the ground reaction force. This is evidence that the center of pressure shift contributes to both force optimization and balance.

### 4.4. Limitations

The number of participants in this study was relatively small. We intentionally excluded statistical estimation and provided common findings for all trials of all participants. However, only two older adults participated in the study. We must be cautious about generalizing our findings to diverse ages.

Several limitations exist in this study due to the simplicity of the human model.

First, muscles other than the tibialis anterior can move the center of pressure backward. According to the general classification, the foot has 52 bilateral bones [4], accounting for one-fourth of the total body. Many muscles connect these bones and work in complex coordination to change the foot shape [34]. However, the tibialis anterior muscles are larger and longer than the intrinsic muscles. As a source of moving the center of pressure, we assumed that it made a greater contribution than the other muscles.

Second, even if we limit our discussion to the tibialis anterior, the muscle runs in three dimensions. For example, the tibialis anterior is attached to the medial side of the foot in the frontal plane. Therefore, its contraction turns the foot inward [34]. However, this study was limited to the sagittal plane. As mentioned previously, the human trunk and limbs are elongated. The sit-to-stand motions were aligned in a straight line from the flexed position when the elongated parts were viewed in the sagittal plane. Therefore, the motion in the sagittal plane was very large.

Finally, the relationship between the center of pressure and mass was verified only in terms of position. As discussed, if velocity information were considered, a clear relationship could have been found. However, this requires extending the argument and performing complex analyses. Clarifying this relationship is a topic for future research.

## 5. Conclusions

The activity of the tibialis anterior contributed significantly to the backward shift in the center of pressure. The center of pressure shift contributed to minimizing the whole-body joint moments. It also contributed to balance as well as force efficiency by adjusting the ground reaction force to be more vertical to achieve motion.

## Figures and Tables

**Figure 1 jfmk-10-00156-f001:**
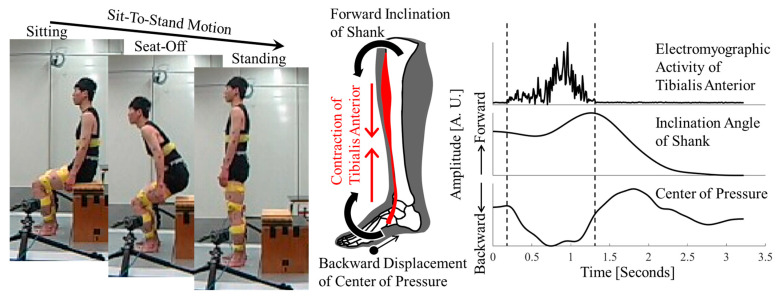
Overview of the study. (**Left**) Measurement scene of sit-to-stand motion. (**Center**) Action direction of the tibialis anterior. (**Right**) Time waveforms of tibialis anterior activity, shank tilt, and center of pressure. This is the actual waveform from the first trial of Participant 1.

**Figure 2 jfmk-10-00156-f002:**
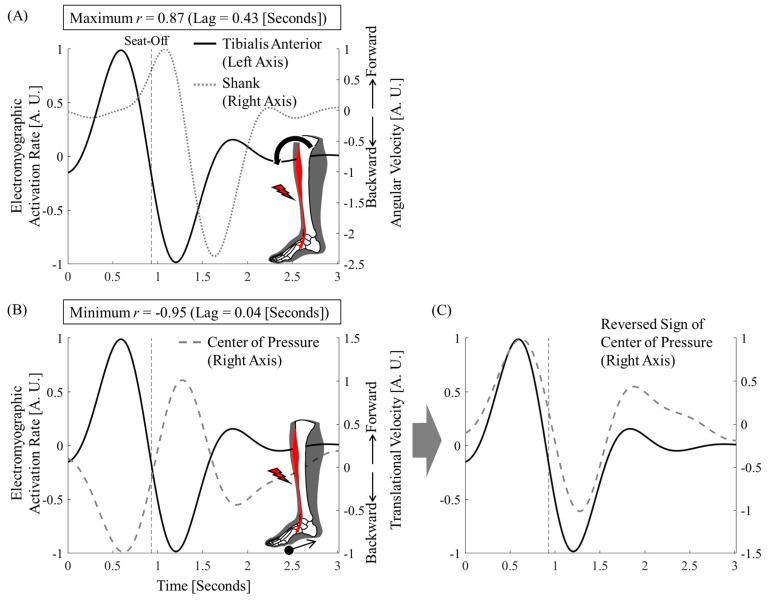
Time-differential waveform of tibialis anterior activity, shank inclination angle, and center of pressure. First trial of Participant 1. (**A**) Time-differential waveforms of tibialis anterior activity and shank inclination angle were positive with a time delay. (**B**) Time-differential waveforms of the center of pressure were negative without a significant time delay (see Figure A1 and Figure A2 for all the participants). (**C**) The sign of the center of pressure is reversed to make it easier to see the synchronization.

**Figure 3 jfmk-10-00156-f003:**
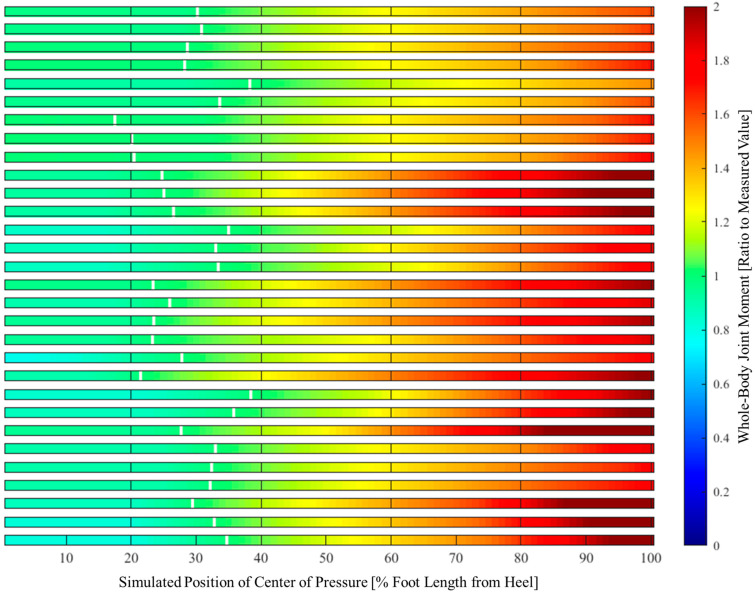
Simulation of whole-body joint moments due to change in the center of pressure. All trials were conducted on all the participants. The last six rows (2 participants × 3 trials) represent the elderly. The white lines represent the measurements. When the center of pressure was forward, the required whole-body joint moment was greater. This was not significantly different from the measurements over a relatively wide area of the back.

**Figure 4 jfmk-10-00156-f004:**
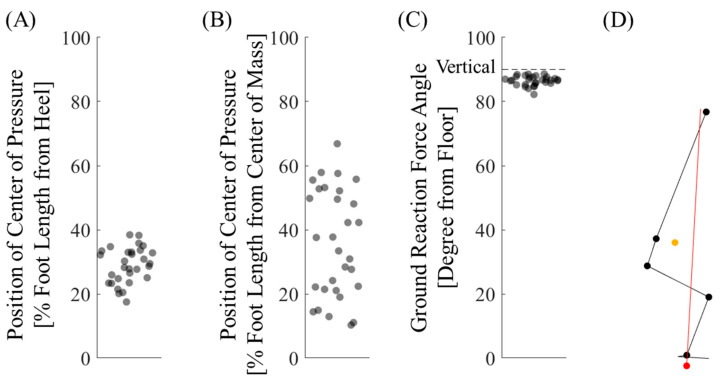
Center of pressure, center of mass, and ground reaction force at seat-off. (**A**) Center of pressure from heel. (**B**) From center of mass. Large variability. (**C**) Ground reaction force angle relative to the floor. The dashed line is vertical. (**D**) Stick figure of the first trial for Participant 1. The center of mass is the orange circle, and the center of pressure is the red circle. The ground reaction force vector is solid red.

**Table 1 jfmk-10-00156-t001:** Participants’ characteristics.

ID	Age (Years)	Height (m)	Weight (kg)
1	30	1.75	60.0
2	22	1.77	68.0
3	22	1.83	65.0
4	27	1.71	54.5
5	21	1.73	60.0
6	30	1.56	52.0
7	21	1.69	57.8
8	22	1.50	58.8
9	71	1.75	57.3
10	67	1.74	73.0

**Table 2 jfmk-10-00156-t002:** Cross-correlation function (maximum/minimum indicates within each trial).

	CCF_TA_Shank_		CCF_TA_CoP_	
	*r* (Maximum)	Lag (s)	*r* (Minimum)	Lag (s)
Mean	0.85	0.37	−0.90	0.13
SD	0.10	0.12	0.05	0.07
Range	0.49–0.97	0.05–0.53	−0.98–−0.80	0.01–0.26

CCF_TA_Shank_: cross-correlation function of tibialis anterior activation rate and shank angular velocity; CCF_TA_CoP_: cross-correlation function of tibialis anterior activation rate and center of pressure translational velocity; *r*: correlation coefficient; SD: standard deviation.

**Table 3 jfmk-10-00156-t003:** Simulated whole-body joint moment (maximum/minimum indicates within each trial).

	Joint Moment (Ratio to Measured Value)	
	Minimum	Maximum
Mean	0.92	1.88
SD	0.06	0.23
Range	0.78–1.00	1.47–2.38

SD: standard deviation.

## Data Availability

The data from this article will be made available by the authors on reasonable request.

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
