# Peer review of "Contribution of Tibialis Anterior in Sit-to-Stand Motion: Implications for Its Role in Shifting the Center of Pressure Backward"

_jfmk, 2025, doi:10.3390/jfmk10020156_

Round 1

Reviewer 1 Report

Comments and Suggestions for Authors

Dear authors, first of all, thank you for submitting your article to JFMK. The article is well structured and follows an appropriate logical flow. The introduction contextualizes the topic well and justifies the relevance of the research. However, some sections could be more concise, especially in the discussion, where there is redundancy in explaining the role of the tibialis anterior in minimizing body load.
Below you can find my detailed comments:
The description of the methods is detailed, however, the sample is only 10 participants (only two of whom were elderly), which limits the generalization of the findings and this should be considered in the text.

The choice of cross-correlation as the main analysis was not sufficiently discussed. In addition, there is no clear explanation of how the data normalization values ​​were defined.

The article mentions that the participant's height was not standardized, but this may influence the biomechanics of the sitting and standing movement.

The results are presented clearly, but some tables could be better organized to facilitate reading. The use of figures and graphs helps interpret the data, but Figure 2 could be more detailed, better specifying the temporal relationship between peaks of muscle activity and displacement of the center of pressure. In addition, I suggest reducing the amount of repetitive text when describing the results, focusing on the main findings.

The discussion reinforces the findings well, but at times there are extrapolations of the results without direct evidence. The comparison with previous studies is useful, but could be expanded, especially for studies investigating muscle activation patterns in the elderly. In addition, the conclusion mentions that the findings may have implications for rehabilitation and training, but does not explore this in detail. I therefore suggest including practical recommendations. 

Author Response

Comment 1: Dear authors, first of all, thank you for submitting your article to JFMK. The article is well structured and follows an appropriate logical flow. The introduction contextualizes the topic well and justifies the relevance of the research. However, some sections could be more concise, especially in the discussion, where there is redundancy in explaining the role of the tibialis anterior in minimizing body load.

Response 1: We thank the reviewer for the insightful comments on our paper. We feel that you have helped us in significantly improving our paper. Based on the reviewers' comments, we have corrected the redundancy, especially in the Discussion. Below are one-by-one responses.

Comment 2: The description of the methods is detailed, however, the sample is only 10 participants (only two of whom were elderly), which limits the generalization of the findings and this should be considered in the text.

Response 2: Thank you for your precise comments. First, we have removed the sentence “to account for differences by age” from the Methods (original manuscript P2L60). Then, we added the following sentence to the Limitations.

(P9L288) The number of participants in this study was relatively small. We intentionally excluded statistical estimation and provided common findings for all trials of all participants. However, only two older adults participated in the study. We must be cautious about generalizing our findings to diverse ages.

              In addition, we have removed previous studies citing findings on the elderly in the discussion (original manuscript P8L230). Other logical leaps in the discussion were also changed in the revised manuscript. Details are provided in Response 7.

Comment 3: The choice of cross-correlation as the main analysis was not sufficiently discussed.

Response 3: We agree with the reviewer. We did not adequately describe the rationale for using cross-correlation. Cross-correlation coefficients are often used to validate motor coordination between body segments. In particular, the lag time at which its correlation coefficient peaks provide a basis for inferring causality from the order of movements. In the revised manuscript, we cited a previous study and added the following sentences.

(P3L103) In this study, cross-correlation coefficients were employed to assess the synchronization between muscle activity and movements. This method enables the evaluation of temporal relationships between the two signals by quantifying the degree and timing of their correlation. Following the approach of the previous study [15], the lag-time corresponding to the peak cross-correlation value was used to determine the temporal offset between the signals. A small lag-time indicates strong temporal synchronization, and a consistent lead-lag pattern provides insight into the sequence of muscle activation and resulting motion.

  • Added Reference: Sasagawa, S., Shinya, M., & Nakazawa, K. (2009). Intersegmental coordination during quiet standing in adults and children: A cross-correlation analysis. Gait & Posture, 29(1), 59–63. https://doi.org/10.1016/j.gaitpost.2008.06.007

Comment 4: In addition, there is no clear explanation of how the data normalization values ​​were defined.

Response 4: We thank the reviewers for their kind advice. The reason for normalizing the amplitude is that the correlation between the magnitude of muscle activity and the resulting movement varies from participant to participant. This is mainly due to the method of measuring muscle activity; different participants have different distances from the measurement location to the source of the potential. We have added a sentence about the reason for normalization along with the cited references.

(P4L124) Each parameter was normalized because the magnitude of the measured muscle activity and the resulting movement would differ between participants. There is a distance from the electrodes to the muscle fibers where the action potentials are detected, and the thickness of the fat layer varies across participants [16].

  • Added Reference: Yang, J. F., & Winter, D. A. (1984). Electromyographic amplitude normalization methods: improving their sensitivity as diagnostic tools in gait analysis. Archives of Physical Medicine and Rehabilitation, 65(9), 517–521.

Comment 5: The article mentions that the participant's height was not standardized, but this may influence the biomechanics of the sitting and standing movement.

Response 5: We greatly appreciate the reviewer's insightful comments. Indeed, participants of different heights exhibit different movement patterns when standing from a platform of the same height. However, it has been found that when the height of the platform is adjusted to match the height of the participant, the movement patterns are generally controlled across participants. In fact, the results of this study showed that the relationship between all trial measurements for all participants was consistent. This is because the proportions of body segment lengths and weights were generally consistent across participants. This explanation was missing from the original manuscript. We have added how we adjusted the measurement environment and the rationale for this.

(P2L70) To eliminate the influence of body height on movement patterns during the sit-to-stand task, seat height was individually adjusted [10]. The seat height corresponded to each participant’s patella height, so that each foot is placed slightly behind the knee joint (Figure 1). This procedure minimizes variability in joint movements and muscle activity between participants, thereby enhancing comparability across participants.

  • Added Reference: Demura, S., & Yamada, T. (2007). Height of chair seat and movement characteristics in sit-to-stand by young and elderly adults. Perceptual and Motor Skills, 104(1), 21–31.

Comment 6: The results are presented clearly, but some tables could be better organized to facilitate reading. The use of figures and graphs helps interpret the data, but Figure 2 could be more detailed, better specifying the temporal relationship between peaks of muscle activity and displacement of the center of pressure.

Response 6: As the reviewer commented, it was difficult to read the notation maximum/minimum because they appeared in the same table with different meanings. Therefore, in the revised manuscript, we annotated the titles of Tables 2 and 3: (maximum/minimum indicates within each trial). Then, we replaced “maximum/minimum” with “range” for all trials in the table.

              For Figure 2, we added a version with the sign of the center of pressure reversed to improve visual clarity of the synchronization.

Comment 7: The discussion reinforces the findings well, but at times there are extrapolations of the results without direct evidence. The comparison with previous studies is useful, but could be expanded, especially for studies investigating muscle activation patterns in the elderly. In addition, the conclusion mentions that the findings may have implications for rehabilitation and training, but does not explore this in detail. I therefore suggest including practical recommendations.

Response 7: We appreciate the very pertinent comments from the reviewer. There were several logical leaps in the discussion of our original manuscript. In particular, we have removed sentences that discuss the relationship between the center of pressure and whole-body load without directly citing experimental results. For example, this included a sentence that discussed the load on each joint in a crouched posture comparing human and bird morphology (original manuscript P8L250). Another example was the consideration of how to maintain consistency between the center of mass, the base of support, and the direction of floor reaction forces at the seat-off (original manuscript P9L293).

              In addition, the reviewer was correct that there were redundant sentences in the original manuscript that repeatedly stated the same facts. In the revised manuscript, we rearranged these sentences so that readers can read the findings concisely. For example, in the original manuscript P7L195, “it was evident that the center of pressure of the foot moved backward during the first half of the sit-to-stand motion “ has been removed because it is followed by similar sentences. Another example was “Balance is defined as a state or phenomenon that remains constant and stable” (original manuscript P9L269)

              Originally, we did not mention implications for rehabilitation. We were uncertain which part of the manuscript the reviewer referred to in their comment. We carefully read the original manuscript and checked for specific references to rehabilitation. We confirmed that the revised manuscript does not include such a section. The purpose of this study was solely to provide basic knowledge of human motion in terms of the relationship between muscle activity and resultant movement.

Reviewer 2 Report

Comments and Suggestions for Authors

The work is very interesting, well articulated, well thought out and well written.
I have no particular criticism to raise except for:
Ten participants is a relatively small number for biomechanics studies, especially considering the division into 8 young and 2 elderly. Two elderly participants alone do not allow reliable conclusions to be drawn on age-related differences.
I suggest explicitly mentioning this limitation in the Discussion or Limitations section, emphasising that the results, especially those concerning age differences (although not explicitly analysed here), call for caution in generalisation.

Author Response

Comment 1: The work is very interesting, well articulated, well thought out and well written. I have no particular criticism to raise except for:

Ten participants is a relatively small number for biomechanics studies, especially considering the division into 8 young and 2 elderly. Two elderly participants alone do not allow reliable conclusions to be drawn on age-related differences.

I suggest explicitly mentioning this limitation in the Discussion or Limitations section, emphasising that the results, especially those concerning age differences (although not explicitly analysed here), call for caution in generalisation.

Response 1: We thank the reviewer for the insightful comments on our paper. We feel that you have helped us in significantly improving our paper. First, we have removed the sentence “to account for differences by age” from the Methods (original manuscript P2L60). Then, we added the following sentence to the Limitations.

(P9L288) The number of participants in this study was relatively small. We intentionally excluded statistical estimation and provided common findings for all trials of all participants. However, only two older adults participated in the study. We must be cautious about generalizing our findings to diverse ages.

              In addition, we have removed previous studies citing findings on the elderly in the discussion (original manuscript P8L230). Other logical leaps in the discussion were also changed in the revised manuscript. Details are provided in Response 7.

Round 2

Reviewer 2 Report

Comments and Suggestions for Authors

no further modifications required